# Overhead Transmission Line Sag Estimation Using the Simple Opto-Mechanical System with Fiber Bragg Gratings—Part 2: Interrogation System

**DOI:** 10.3390/s20092652

**Published:** 2020-05-06

**Authors:** Krzysztof Skorupski, Damian Harasim, Patryk Panas, Sławomir Cięszczyk, Piotr Kisała, Piotr Kacejko, Janusz Mroczka, Michał Wydra

**Affiliations:** 1Department of Electronics and Information Technology, Lublin University of Technology, 20-618 Lublin, Poland; k.skorupski@pollub.pl (K.S.); d.harasim@pollub.pl (D.H.); p.panas@pollub.pl (P.P.); s.cieszczyk@pollub.pl (S.C.); 2Department of Power Systems, Lublin University of Technology, 20-618 Lublin, Poland; p.kacejko@pollub.pl (P.K.); m.wydra@pollub.pl (M.W.); 3Electronic and Photonic Metrology, Wroclaw University of Technology, 50-317 Wroclaw, Poland; janusz.mroczka@pwr.wroc.pl

**Keywords:** power system monitoring, power lines sag measurement, FBG sensors interrogation, strain measurement

## Abstract

This article presents the use of a sensor with fiber Bragg grating along with an interrogation system used for monitoring the overhead lines’ wire elongation. The possible interrogation methods based on adjusted filters were considered. In the experimental part, three types of fiber Bragg grating pairs, characterized by a small shift in spectra in pairs and gratings with exact matching, were examined. The study showed that, by choosing the appropriate mechanical parameters of the elongation transformer with the optical parameters of the sensor and dedicated filter, the optomechanical system can be adjusted to the required range of overhead line wire sag observation. The range of sag depends on the distance between the poles, the wire type, and its real length in the span, which effectively determines the sag.

## 1. Introduction

Currently, electricity is necessary for the functioning of large enterprises and urban agglomerations, as well as municipal recipients living both in cities and the countryside. Electricity companies are required to provide an uninterrupted power supply with the normative power quality parameters [1,2]. Satisfying the ever-higher expectations of electricity consumers can only be met by ensuring the required reliability, management, and control of all elements of the power system, from generating units (power plants), through the power transmission network, to the distribution network and receiving installations.

The importance of power line wire elongation measurement is connected with sag calculation, which is the critical parameter of high voltage power line safe operation. The sag strictly depends on wire temperature, which is a function of atmospheric conditions and load current. At present, there are commercially available technologies that enable the calculation of the sag value [3,4]. Typical wire sag calculations cover two stages, where the first stage is a determination of wire temperature using thermal models, weather, and current measurements [5,6], and the second stage assumes mechanical sag-tension calculations based on power line technical parameters [7]. There are also systems that combine the weather and tension measurements [3] or camera-based vision systems [8]. On the basis of the sag value, it is possible to determine the dynamic line rating and control power line operational safety. The presented method simplifies the sag calculations by direct and accurate measurements of power line wire elongation using the system based on fiber Bragg gratings.

This article presents a photonic measuring system using simple homogeneous fiber Bragg gratings (FBG), enabling indirect temperature and wire tension measurements based on the optical signal. This facilitates the determination of the maximal allowable load current of the monitored power line section. Increasing energy efficiency is one of the priorities of energy policy. Many countries are required to implement numerous measures aimed at increasing the efficiency of energy production, distribution, and end-use. Most of the papers dealing with interrogation systems are mainly focused on the properties of individual systems such as linearity and sensitivity. In fact, the interrogator system is based on known solutions, but it has never been used for such physical facilities and never been presented in the form of a ready to use device that informs the user about the value of power transmission line sag. An element of novelty is to demonstrate the impact of fiber Bragg gratings spectral characteristics selection on the transition characteristics and indication of optimal system. In this paper, we present a complex measuring system. The stress and sag of a power transmission line is converted into the elongation of the control plate through the mechanical system. 

Bragg gratings can be used to measure various mechanical quantities such as strain [9], rotation and a twist [10,11], curvature [12], or bending [13]. There are many signal interrogation concepts from fiber Bragg grating transducers [14,15,16]. One of the proposed methods is to use an insulated Bragg grating, whose central wavelength can be changed in a controlled manner. The basic operation in such a technique is to use the interrogation grating as a barrier filter (band-stop filter or band-rejection filter) that strongly reflects the light returning from the Bragg sensor. To do this, it is necessary to create two gratings with the same central Bragg wavelength value. The wavelength shift of the reflected light from the measuring FBG element, caused by its deformation, can be tracked by the receiving barrier filter (band stop/band rejection). This is done by the controlled deformation of the filter grating using, i.e., piezoelectric actuators until the Bragg wavelengths of both gratings overlap [17,18]. Using this technique, it is possible to use multiple Bragg sensors on a single fiber. In such a case, it is necessary to create a barrier filter consisting of FBGs with identical central Bragg’s wavelengths as the gratings used as sensors, with each filter grating being independently tuned. The system developed in that way makes it possible to find the wavelength of light reflected from the corresponding measuring element [19].

## 2. Existing Solutions

The conventional, uniform fiber Bragg gratings (FBG), in comparison to modified, untypical fiber periodic structures, are easy to inscribe and do not require complicated diffractive elements in laser beam transmitting systems such as nonuniform or apodized phase masks. Using the phase mask method for writing gratings with a modified period requires the application of diffraction masks with a variable period of its structure. In addition, structures with tilted zones of refractive index modulation need precise control of the rotation angle between the fiber and interference pattern induced by diffractive elements. The response optical spectrum shape analysis of FBGs, applied as the physical quantities of transducers, causes difficulties because of the optical spectrum analyzer or spectrophotometer’s requirements. Such devices are expensive, not portable, and not capable of work in external, adverse environmental conditions. The idea of sag estimation using the experimental setup with nonuniform period gratings was presented in a previous paper [20], where it was proposed that uniform Bragg gratings could be utilized as both elongation sensors and optical filtering structures, which are the elements of the interrogation system used for the optical measurement circuit for overhead transmission power line sag estimation. 

In existing measurement systems, the measured quantity (i.e., the temperature or tension) usually causes a shift in central Bragg wavelength of FBG sensors. This is the basic advantage of fiber optic gratings used as sensors: The information is carried by wavelength shift and not in the radiation intensity. This makes the measuring system with such a sensor resistant to changes in transmission losses of the measurement circuit and to changes in the intensity of the source radiation.

Unfortunately, determining the Bragg wavelength shift requires expensive and slow operating devices, such as spectrum analyzers, or is based on methods that transform the change in the Bragg wavelength into radiation intensity. The simplest demodulation system is based on a matched filter method, in which the interrogating grating has the same spectral shape as the sensor grating. Generally, such a system can work in reflection or transmission mode. In reflection mode, the radiation reflected from one FBG (sensor) is transmitted to the other grating, which also reflects the radiation that is then measured on the detector [21]. 

When wavelengths’ centers of gratings are more shifted in relation to each other, less radiation reaches the detector. In the optical transmission system, only one (sensor) FBG reflects the radiation, and the interrogating grating is a filter that blocks this radiation [22,23]. As a result, when the Bragg wavelengths’ centers of gratings are matched, the least radiation reaches the detector. When the sensing grating wavelength is shifted, the radiation power that reaches the detector increases. In both cases, the so-called filter matching means that the spectral characteristics of both gratings are identical, or at least have a similar spectral width [24]. The use of chirped gratings in such simple systems allows for an extension of the measuring range and additionally improves the linearity of the transient characteristic [25,26].

There are also systems in which the filtering component has a much wider spectrum than the sensor FBG, and it can also be the spectrum of the SLED (super-luminescent emitting diode) radiation source itself [27]. The radiation spectrum of a source with a much narrower full width at half maximum (FWHM) than a sensor grating, such as a laser, can also be used.

This article presents the application of a sensor with a Bragg grating with an interrogating system to monitor the elongation of the overhead power transmission line wires. Several interrogator systems based on matched filters were considered. There are interrogator solutions that can operate in variable environmental conditions, e.g., when measuring currents in power lines [28]. In this project, we decided to develop an interrogator following our design with electric current surge protection and that enables measurement with minimized power consumption. Different pairs of FBG gratings were used, each characterized by a small shift in spectra in individual pairs and gratings with precise matching. This paper shows that choosing the mechanical parameters of the elongation transformer and the parameters of the optical sensor and filter system allows the opto-mechanical system to be adapted to the required range of sag estimation.

In fact, there are portable interrogating devices which can operate in variable environmental conditions. However, their properties make it difficult to use this type of interrogator when measuring parameters of power transmission line. First, the operating temperature range of market available interrogator is from −20 °C to 64 °C, which is insufficient for measuring sag of real existing power lines. In real conditions the temperature of wire conductor is changing in wider range. Presented interrogating device has integrated temperature controller which allows it to work in the temperature range from –40 °C to 180 °C. In addition, application on power transmission line requires that the device will be compact and the system weights as least as possible. Another advantage of designing device is energy consumption optimization, which allows for operation using battery power. The cost of presented solution is approximately four times lower than the market price available. This is mainly due to the fact that the system is dedicated for a specific application—for interrogating Bragg sensors mounted on power transmission lines.

## 3. Materials and Methods

This paper presents the research and development of the interrogation system, whose task it is to enable the conversion of changes in the optical parameters of the photonic sensor (fiber Bragg grating) spectrum to changes in radiation power. By using optoelectronic components such as photodetectors, changes in optical power can be converted to changes in DC voltage. A voltage value in analog form is commonly used as an information carrier and can be easily converted into a digital value that can be subjected to further numerical processing. It was assumed that the designed detection system must be characterized by high accuracy and ensure the possibility of covering a given measuring range matched to the expected changes in the power transmission line tension. This is possible due to the appropriate design of the detection system and the appropriate selection of optical filter spectral parameters relative to FBG sensors. 

The interrogator is a part of the designed complex system, which will measure both elongation and temperature of power line. This will allow for the detection of wire sag occurring during the operation in conditions of catastrophic rime and power line overload. The use of both temperature and conductor elongation measurements will ensure the redundancy of received results. Two options were considered when designing the solution. Variant 1 assumes attaching both FBG sensor and FBG filter near the power line conductor (Figure 1). 

Solution presented on Figure 1 will be characterized by insensitivity for temperature changes. The system will measure elongation of power line wire. Variant 2 provides attaching FBG sensor on the power line and FBG filter in the thermostated housing placed on transmission tower (Figure 2).

In the second variant, the output signal of the system will consist of summarized spectral shift caused by power line temperature and elongation changes. This work presents the operation of a system module responsible for measuring the line elongation. The complex system will provide measurements of temperature and elongation using two separated interrogation systems, so we decided to separate sensing and filtering FBGs. Following this assumption, the system part intended for elongation measurement measures both temperature change and elongation. The power line extension will be determined as the difference between results of elongation measuring module and temperature measuring module. As it is mentioned before, the system will also contain a temperature measurement module based on the FBG interrogator. Temperature measurement requires maintaining a stable constant temperature for the FBG filter or its control. This is difficult to implement on the transmission line wire, so it was decided to place filter grating on the power line support tower. The measurements results presented in this manuscript were carried out using the variant 2 system while considering influence of temperature changes on the operation of system. 

The idea of the measurement method is shown in Figure 3, where three types of variant 2 interrogation systems for power line sag sensors, based on FBG transducers, are collated for comparison. First of all, various combinations of the systems for interrogation of a single FBG operating as a sensor were analyzed. 

It is worth noting that in each of these systems, a different Bragg grating works as a band-stop filter matched with both the central Bragg wavelength and the FWHM parameter, which describes the spectral width. In the first case (Figure 3a), the FBG sensor, which is affected by a sag change of the power transmission wire, is located just before the photodetector, which measures the signal reflected from the optical filter. In the second case (Figure 3b), the sensor is placed at the end of the measuring fiber, and the reflected signal is transmitted via an optical circulator and is then filtered by another FBG located before the photodetector. In the third case (Figure 3c), the sensor is placed at the end of the measuring fiber (the same as the second case), but the optical filter is located in the light source arm, directly after the SLED.

The overhead transmission line wire sag measurement schemes are shown in Figure 3. The presented systems facilitate strain and sag measurement. The light source was a superluminescent light-emitting diode (SLED), S5FC1005S Thorlabs (Newton, NJ, USA). The interrogation systems shown schematically in Figure 3 are characterized by high simplicity of design. Both use an optical circulator to selectively transmit an optical signal from channels 1 to 2 and from channels 2 to 3. Good class circulators are characterized by the selectivity of unwanted transmission between channels at 50 dB, which makes the transmission effect from channels 1 to 3 negligible.

In the case of the system shown in Figure 3a, a significant drawback is that the photodetector is measuring the optical power of the signal transmitted through the sensing structure. As a result, it is necessary to complicate the sensor head itself and provide two fibers: one supplies the radiation that illuminates the sensor, and the second transmits the signal from the sensor to the detector. The systems presented in Figure 3b,c do not have this drawback. They offer so-called single-end operation, where the signal reflected from the grating operating as the transducer of the measured quantity is taken into account. The proximity of the grating working as a filter to the light source or photodetector is advantageous due to the need to maintain each of these elements’ constant operating temperature.

The measuring systems described in Figure 3a,b were chosen to test for power transmission line sag measurement. In order to verify the correctness of interrogation, measurements recording the output signal were performed using two methods: the first used an optical spectrum analyzer, and the second used an optical power detector.

In the first case, the results are spectral characteristics showing changes in the optical signal caused by the strain of the Bragg grating used as a sensor. However, in the following case, the results of the study are transient characteristics. Figure 4 schematically shows the experimental stand used to measure power line wire sag using the actual power transmission line.

To enable the sensor to be mounted on the power transmission line wire, a special elongation transformer was designed (Figure 5), allowing the transformation of the power line sag to extend the FBG sensor.

One of the most important features of the system is that it is mounted non-invasively, directly on the monitored span of the line. The control plate has an additionally shaped measuring section with non-parallel edges, on which, after attaching the photosensitive single-mode optical fiber with the inscribed Bragg grating, an elongation sensor is created. The length variations of the conductor in the measured section of the power line imply the reaction of a photosensitive single-mode fiber with Bragg grating, leading to a shift in the spectral characteristics of the reflected light signal. The change in the characteristic shift is not affected by electromagnetic field changes. The design of the elongation transformer ensures the stability of its elements mounted on the conductor without any negative effects on its strength, mainly due to the non-invasive method of attachment. In case of any damage to delicate components of the sensor system, it is possible to replace them with new ones with the same parameters.

## 4. Results

We begin by detailing the installation of the entire elongation transformer system with the control plate and FBG sensor on the experimental outdoor test stand, which was used for the simulation of power line conductor operational conditions. The system was mounted on an ACSR Hawk wire (FPE, Będzin, Poland) during tests loaded with AC electric current controlled within a range of 0–1020 A. The Bragg gratings used were inscribed on fibers hydrogenated for a period of seven days at a pressure of 120 bar and a temperature 20 °C. In order to verify the influence of the filter spectral characteristic shape on the sensor properties, three identical FBGs were used as interrogating elements. They are referred to in the following sections collectively as FBG sensor. Three filters with different spectral characteristics were also made and are referred to as fiber Bragg grating filter 1 (FBGF1), FBGF2, and FBGF3, respectively. Figure 6 presents the spectral characteristics of all three pairs of gratings used.

Figure 7 presents the spectral characteristics measured using an optical spectrum analyzer and transient characteristics of the system that allow observation of the value of voltage generated by the photodetector system. The presented spectra in Figure 7 were measured by OSA with the corresponding transient characteristics obtained using a photodetector and the investigated interrogation method.

Measurements were performed using three pairs of periodic structures, indicated in Figure 6 as pair 1 (Figure 6a), pair 2 (Figure 6b), and pair 3 (Figure 6c). Bragg gratings were arranged in pairs differing in spectral characteristics of the filters, which had different values of Bragg wavelengths in order to show the impact on the measurement system. Figure 7 shows the results of spectral measurements and optical power measurements after passing through the interrogation FBG in the system shown in Figure 3a.

For comparison, Figure 8 also shows the characteristics made with the FBG + FBGF1 matched pair but using the interrogation system in Figure 3b. The nature of the changes in the optical spectrum is similar, but there are differences in the spectral characteristics of the sensor and filter. Therefore, the processing characteristics are different. The most important problem here is a quite narrow range of voltage changes on the photodetector. In the case of the systems used, the normalized value of voltage varies in the range of 0.67–1.00 V in the case of the interrogator in Figure 3a and 0.675–1.00 V in the case of the interrogator in Figure 3b. 

As presented, the change in the interrogation system configuration affects the processing characteristics. In order to extend the range of voltage changes on the photodetector, it was decided to reconfigure the detection system with an additional optical circulator. The entire interrogator system is shown in Figure 9. It is worth noting that, in the presented case, both the sensor and the filter gratings are placed in separate arms of the interrogator. 

In the case of the systems shown in Figure 3a,b, it can be seen that, despite the adjustment of the central Bragg wavelengths of both periodic structures, the minimal power level recorded by the detector is about 60%–70% of the maximum. This means that there is a reduction in the system response dynamics. The solution to this problem is the use of the system in which the signal reflected from the sensor grating is directed via a second circulator to the filter in a band-pass configuration. As a result, the photonic structures used in the developed system do not need to have reflectance at a level close to 100%. This configuration is shown in Figure 9. The structure of the system shown schematically in Figure 9 causes the common part of the signal reflected from both periodic structures to be directed to a photodetector. Thanks to this, it is possible to obtain the maximum value of optical power registered by the light detector when matching the Bragg wavelength of both gratings, and the minimal value of generated voltage is about 20% of the maximum. To set up the system presented in Figure 9, an SLED light source was equipped with temperature controllers and a current flowing through the THORLABS S5FC1550P-A2 diode and an optical power detector THORLABS PDA30B-EC. The spectral characteristics of the output signal were measured using the AQ6370D Yokogawa Electric Corporation (Tokyo, Japan) optical spectrum analyzer. The interrogator system is shown in Figure 10. 

For the system shown in Figure 9, spectral measurements were also carried out using an elongation transformer on a real overhead power transmission line wire. The results of these measurements are presented in Figure 11. It can be observed that there is a clear extension of the normalized range of voltages measured by the photodetector, from 0.21 V to 1.00 V. It can also be noted that the nature of voltage changes due to the increasing tension of the power line conductor is the same for all results presented in Figure 7, Figure 8, and Figure 11 because it does not depend on the interrogator itself but on the design of the elongation transformer.

As mentioned earlier, to demonstrate the influence of the difference between the optical characteristics of the filter and sensor, measurements were performed for three pairs of Bragg gratings. Figure 12, Figure 13 and Figure 14 present the results of measurements of the power line stress for the FBG + FBGF2 matched gratings pair.

Figure 15, Figure 16 and Figure 17 show the results of measurements of the power transmission line conductor stress for the FBG + FBG3 matched gratings pair.

Based on the presented analysis of spectral characteristics, it can be seen that the system with two optical circulators and a Bragg filter in a bandpass configuration is characterized by clearly increased response dynamics. The transient characteristics of the system are in the ranges 4.4–6.6 V, 4.4–7.0 V, and 1.8–6.4 V for the cases depicted in Figure 15, Figure 16 and Figure 17, respectively. The useful range of voltages generated at the output is, therefore, twice as large in comparison to interrogation setups with one circulator. Practical verification of the proposed photonic system operation has been carried out at an outdoor test stand, constituting a 50 m span with a suspended ACSR Hawk cable (240 mm^2^) with the possibility of forcing currents in the conductor within the range of 0–1020 A. The scheme of the test stand is shown in Figure 18, where the main elements of the test stand and the three most important measuring systems have been marked: the system for measuring horizontal wire tension using a Instron PM-L 2526-802 10 kN transducer (Norwood, MA, USA), with an accuracy of 0.5%; the system for measuring the sag change using a Micro Epsilon WDS 96/2500 mm (Raleigh, NC, USA) transducer, with an accuracy of ± 2.5 mm; and the system used to measure the surface temperature of the conductor based on the temperature sensor attached to the ACSR Hawk wire with thermally conductive glue [8].

The developed outdoor test stand is presented in Figure 19 and its key elements are indicated.

## 5. Discussion

Section 4 presents the results of measurements made on an outside test stand with an installed ACSR Hawk wire during tests loaded with an electrical current. The results of the photodetector voltage dependence on wire stress and wavelength shift were also presented. However, from the applicability point of view of the proposed system, it is important to demonstrate the possibility to measure the power line’s wire sag. This section presents the discussion of how photodetector voltage can be a measure of power line sag. The signal processing characteristics of the entire measurement system have also been presented. During tests, the proposed photonic measuring system in this paper was installed on the ACSR Hawk conductor. The wire tension, temperature, and sag were recorded by the dedicated measuring reference system on the developed outdoor testbed presented in the previous section. The installation in the testbed with a controlled high-current source allowed for loading of the ACSR Hawk conductor with 1020 A, which caused the wire to heat up to 60 °C and resulted in its elongation. Measurements were made on high voltage power line wire with 50 m span. By using a mathematical model of power line conductor shape, the results were also calculated for other exemplary wire spans. Table 1 summarizes the characteristic parameters of examined and measured spans.

The schematic in Figure 20 presents the parameters for all of the characteristics of the tested cases.

The relationship determining the sag *D* value of an overhead line wire in the span of length *S,* and actual conductor length *L*, can be described with Equation (1) [29]: (1)L=(2H/w)[sinh(Sw/2H)].

It is common to perform sag-tension calculations using only the horizontal tension component *H*. The conductor length in the span can be calculated with the application of a catenary equation using Equation (1). The right side of Equation (2) represents the parabolic approximation of the catenary equation. The hyperbolic sine in Equation (1) can be replaced by a simplified form, and then the length of the wire can be described as presented in Equation (2):(2)$$L=S\left[1+S^{2}w^{2}/24H^{2}\right].$$

However, taking into account sag *D,* the total length of the conductor *L* can be expressed according to Equation (3):(3)$$L=S+\left[8D^{2}/3S\right].$$

A simple transformation of Equation (3) allows the determination of the sag *D* as a function of the distance between the poles *S* and the actual length of the wire *L*, which can be expressed by Equation (4):(4)$$D=\sqrt{\left[3S/8\left(L-S\right)\right]}.$$

The practical verification of the metrological properties of the photonic strain sensor has been performed on real power lines, and the parameters of this are summarized in Table 1. 

Long-term measurements of the horizontal tension force, sag, and temperature of the ACSR 27/6 Hawk conductor at the outdoor test stand were performed. During load tests, the atmospheric conditions and electric current were recorded. Figure 21, Figure 22 and Figure 23 summarize the values of the power line sag as a function of the voltages measured by the photodetector for three selected power line spans that differed in values of the distance *S* between the poles, within the associated lengths *L* of the conductor, according to Table 1.

The results presented in the previous section proved that the most favorable interrogator system from the metrological parameters point of view is the system shown in Figure 9; all indirect measurement results have been presented for this particular system. For comparison, the results have been calculated for three pairs of FBG structures (pair 1: FBGF1 + FBG sensor; pair 2: FBGF2 + FBG sensor; pair 3: FBGF3 + FBG sensor).

As mentioned above, the measurements were made on three power line spans, but for all three, the increases in the range of voltage changes measured by the photodetector were visible. Loading a line with a length of 50 m with an electric current varying from 0 to 1000 A caused a voltage change of 3.26 V and 3.21 V in the case of the FBGF1 + FBG sensor pair (Figure 6a) and the FBGF2 + FBG sensor pair (Figure 6b). The use of the pair of the interrogator system in the form of the FBG3 + FBG sensor pair (Figure 6c) shows that, with the same sag changes, the range of voltage change on the photodetector is about 25% greater, at 4.52 V. The situation is similar for measurements made on a line with a span of 150 m. For the FBG #1 and FBG #2 pairs, the voltage changed from 3.15 V and 3.25 V but also increased to 4.52 V for the FBG #3 pair. The same effect can be observed in the case of measurements made for a 300 m span line, where the use of a filter-sensor pair with the most matched spectral characteristics results in an extension of the measured voltage went from 3.25 V to 3.34 V for FBG #1 and FBG #2, respectively, up to 4.52 V for FBG #3 pair.

The results shown in Figure 21, Figure 22 and Figure 23 reveal that, in the case of power lines’ flat spans, regardless of the distance between the poles *S* in the span, there is a visible increase in the voltage change measured by the photodetector. The electric current conductor loading on 50 m, 150 m, and 300 m spans caused changes in voltage measured by the photodetector after passing through the demodulation system in the form of an interrogator. It is therefore possible to use a sensor with a Bragg grating with an interrogation grating to monitor the elongation and sag of an overhead line’s span. In the presented paper, several interrogator systems based on matched filters have been examined and analyzed, and the tests have been verified for various pairs of FBG gratings characterized by a small spectral shift in individual pairs.

During measurements at the experimental stand for simulation of power line conductor operation, temperature changes were forced in the range of 20–80 °C. That caused the extension of tested wire from 48.293 m to 48.347 m, which corresponded to a 0.369 m sag change. The determination error was within 5%. It should be added that the sag was measured (as a reference) by a draw wire displacement sensor, which uses tensometric strain transducers. This kind of gauge is sensitive to electromagnetic interference, and its application in power line conditions measurement could lead to incorrect results.

It seems that at higher temperatures causing greater sag value the error may increase even above 10% and 15%. However, this kind of information is very valuable because it is not available by other methods. In addition, such critical sag values are extremely rare. The system operator only needs information about their occurrence. The increase of error results from the transition nonlinearity of the sensor and interrogating filter.

## 6. Conclusions

This work, which involved the direct application of photonic sensors in electrical power engineering, led to an increased efficiency of energy transmission and an improvement of its reliability. The results presented in this work may be a response to the demand for intelligent and effective management of the power grid. The developed system for power line sag monitoring proposed in this article may be useful for transmission and distribution line maintenance, reducing transmission losses with appropriate control, increasing network efficiency and achieving higher energy efficiency, and reduced investments in power infrastructure and large-scale energy sources.

The work presents the use of a sensor with a fiber Bragg grating together with an interrogation system to monitor the elongation and sag of overhead transmission lines. Possible interrogator systems based on matched filters were considered. In the experimental part, three different FBG grid pairs were used, characterized by a small shift of spectra in pairs (two types) and gratings with an exact matching. The transient characteristics of individual grating pairs were determined. The voltage level changes on the detector were analyzed for two versions of interrogators using a single circulator. The system with two circulators was proposed; this system exhibited the greatest changes in detector voltage for the tested range of stresses. For individual pairs of gratings, the transient characteristics were found to have a linear response associated with the main Bragg peak and a non-linear range having a saturation nature associated with sidebands from the shorter side. The next part used the proposed measuring systems to determine the relationship between the voltage values and the detector and the conductor sag. A set of such characteristics were determined experimentally for three types of grating pairs and for three distances between the transmission line poles. 

The study showed that by choosing the mechanical parameters of the elongation transformer and the optical parameters of the system (the sensor and filter), the optomechanical system can be adapted to the required range of sag observation. The range of sag depends on the distance between the poles and the effective length of the power line wire. The shape of the gratings, in turn, can affect the transient characteristics, allowing the sensor system to operate in a linear range for a typical range of expected sags. The nonlinear range, on the other hand, allows the measurements of larger than typical sag values with less sensitivity associated with a logarithmic type of nonlinearity.

In the future, the authors are planning to develop a sensor head ventilated cover and investigate its influence on sensor measurement accuracy. The presented results are part of a project aimed at implementing the device for measurement of power transmission line temperature and elongation. The solution is currently validated at 6 TRL. The results of measurements performed on the simulation stand allow us to commence the next stages of project. The following levels include integration of optical fiber modules and designed optoelectronic devices. This work is carried out with a company related to the power energy industry, so it will enable measurements to be conducted under real conditions. The final stage of concept development will be the integration of the device with the high-voltage transmission line management system.

## Figures and Tables

**Figure 1 sensors-20-02652-f001:**
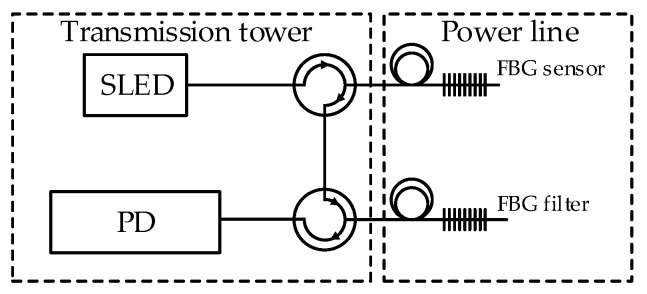
Interrogation system variant 1. SLED: superluminescent light-emitting diode; PD: photodetector.

**Figure 2 sensors-20-02652-f002:**
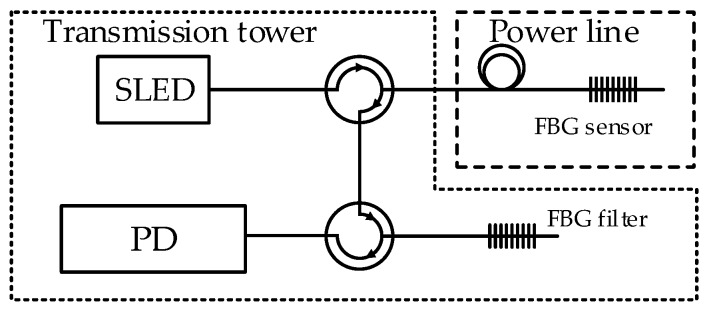
Interrogation system variant 2.

**Figure 3 sensors-20-02652-f003:**
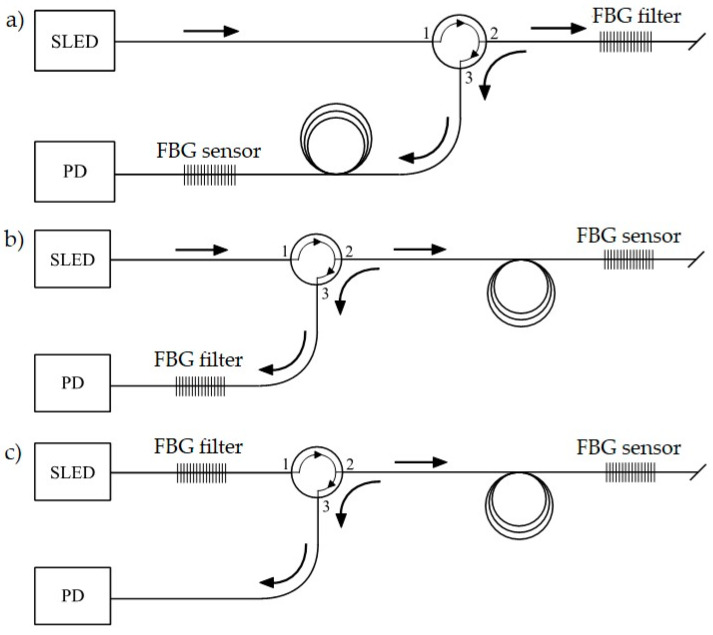
The interrogation system for the overhead transmission line wire sag measurement: (**a**) Sensor located in the photodetector arm; (**b**) sensor located in the sensing arm; (**c**) sensor located in the input arm. SLED: superluminescent light-emitting diode; PD: photodetector.

**Figure 4 sensors-20-02652-f004:**
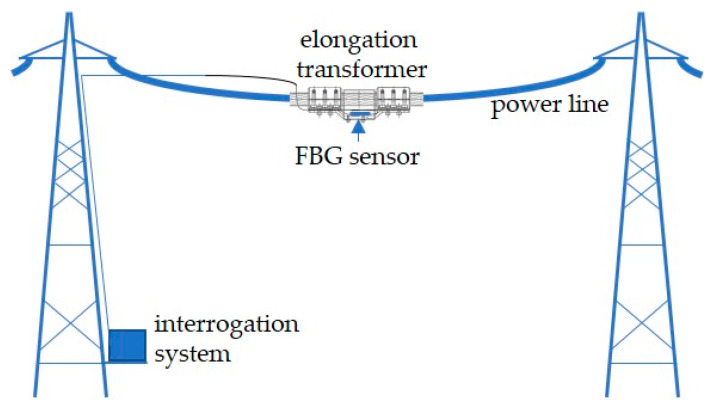
The overhead transmission line wire sag measurement system.

**Figure 5 sensors-20-02652-f005:**
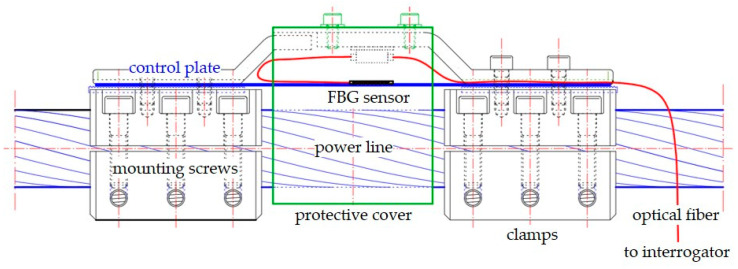
The elongation transformer used in tests of sag measurement.

**Figure 6 sensors-20-02652-f006:**
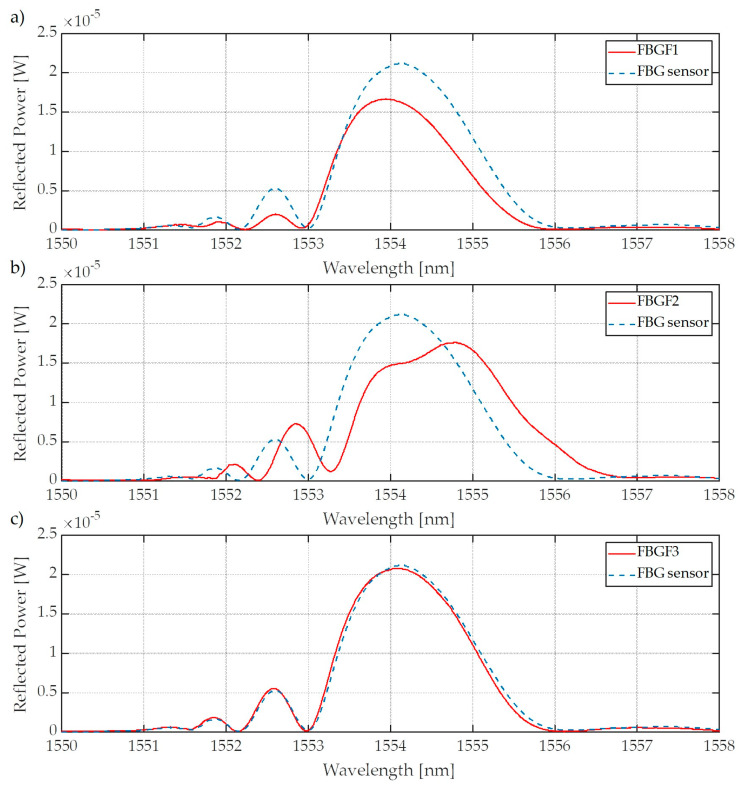
The spectral characteristics of gratings used in measurements: (**a**) filter Bragg grating (FBG) filter with a center wavelength shorter than the sensor, (**b**) FBG filter with a center wavelength longer than the sensor, (**c**) FBG filter with a center wavelength equal to the sensor.

**Figure 7 sensors-20-02652-f007:**
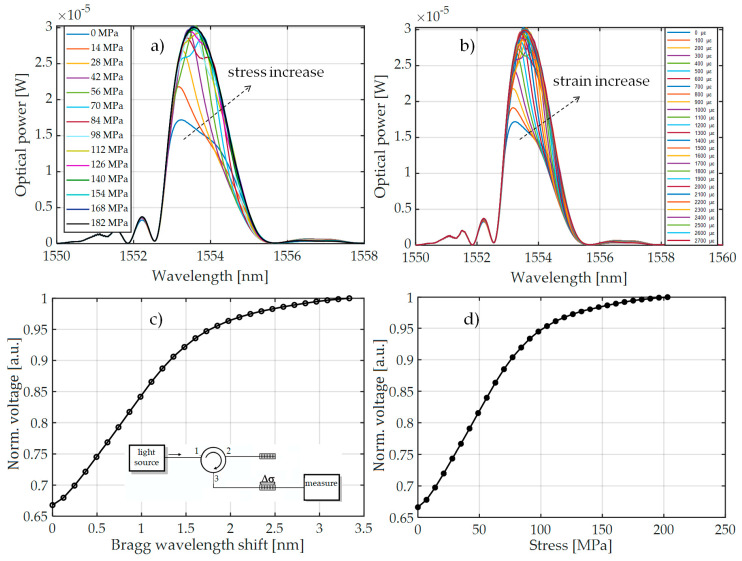
The measurement results obtained from the interrogation stand (as in Figure 3a) for the matched grating pair FBG + FBG1: (**a**) Spectral response change for an increase of tension; (**b**) spectral response change to an increase of strain; (**c**) the transient characteristic of wavelength shift to direct current (DC) voltage generated by the photodetector; (**d**) characteristics of power line wire tension conversion to a change of photodetector voltage.

**Figure 8 sensors-20-02652-f008:**
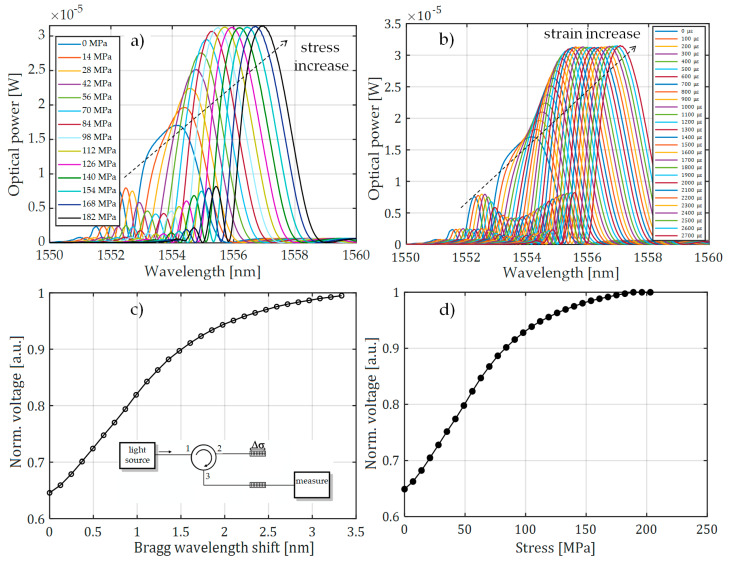
The measurement results obtained from the interrogation stand (as in Figure 3b) for the matched grating pair FBG + FBG1: (**a**) spectral response change for an increase of tension; (**b**) spectral response change for an increase of strain; (**c**) the transient characteristic of wavelength shift to DC voltage generated by the photodetector; (**d**) characteristics of the power line wire tension conversion to a change of photodetector voltage.

**Figure 9 sensors-20-02652-f009:**
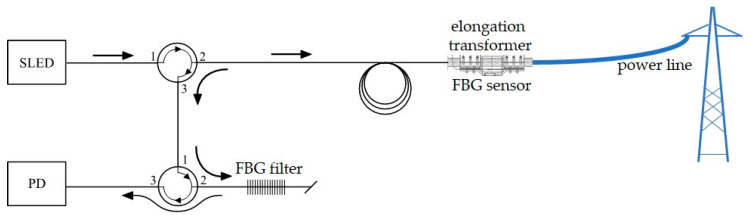
The final interrogation system for the overhead transmission line wire sag measurement. SLED: superluminescent light-emitting diode; PD: photodetector.

**Figure 10 sensors-20-02652-f010:**
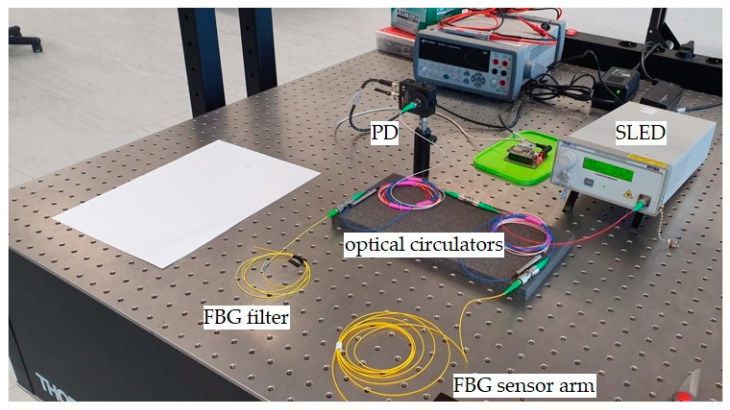
Laboratory version of the interrogator used for overhead transmission line wire sag measurement. SLED: superluminescent light-emitting diode; PD: photodetector.

**Figure 11 sensors-20-02652-f011:**
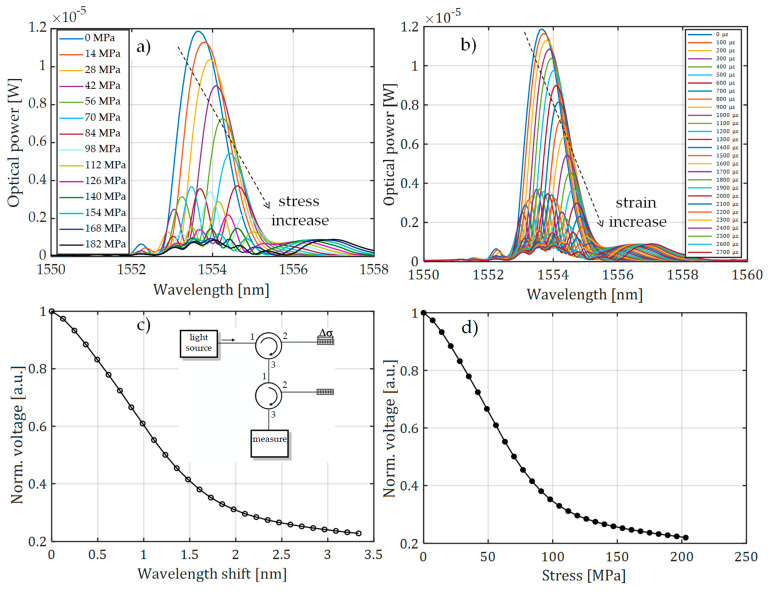
The measurement results obtained from the interrogation stand (as in Figure 9) for the matched grating pair FBG + FBG1: (**a**) spectral response change in response to an increase of tension; (**b**) spectral response change to an increase of strain; (**c**) the transient characteristic of wavelength shift to DC voltage generated by the photodetector; (**d**) characteristics of power line wire tension conversion to a change of photodetector voltage.

**Figure 12 sensors-20-02652-f012:**
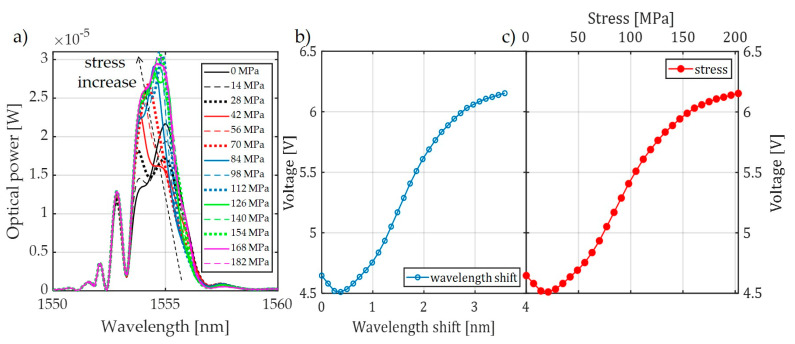
Measurement results in the interrogator system showed in Figure 3a for the FBG + FBGF2 matched gratings pair: (**a**) change in the shape of the system output signal due to stress change; (**b**) conversion characteristics of the wavelength shift into a voltage change measured on the photodetector; (**c**) the characteristics of the power line conductor stress conversion to change the voltage on the photodetector.

**Figure 13 sensors-20-02652-f013:**
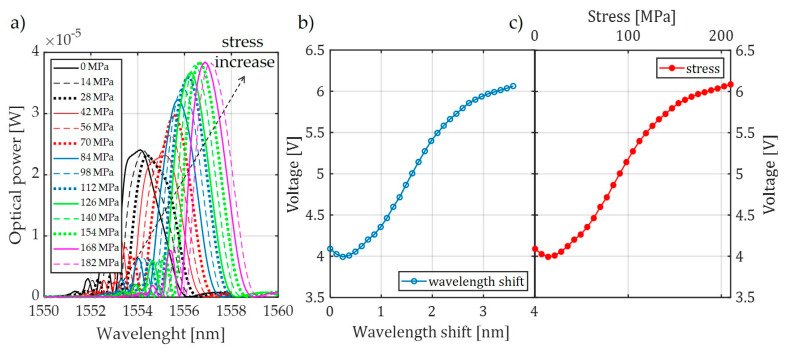
Measurement results in the interrogator system showed in Figure 3b for the FBG + FBGF2 matched gratings pair: (**a**) change in the shape of the system output signal due to stress change; (**b**) conversion characteristics of the wavelength shift into a voltage change measured on the photodetector; (**c**) characteristics of the power line conductor stress conversion to changes in the voltage on the photodetector.

**Figure 14 sensors-20-02652-f014:**
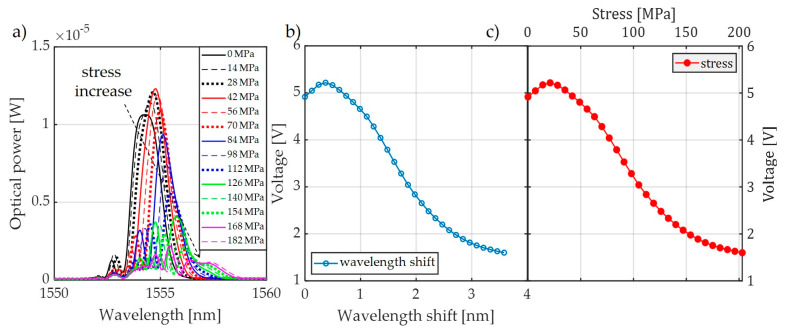
Measurement results in the interrogator system shown in Figure 9 for the FBG + FBGF2 matched gratings pair: (**a**) change in the shape of the system output signal due to stress change; (**b**) conversion characteristics of the wavelength shift into a voltage change measured on the photodetector; (**c**) the characteristics of the power line conductor stress conversion to changes in the voltage on the photodetector.

**Figure 15 sensors-20-02652-f015:**
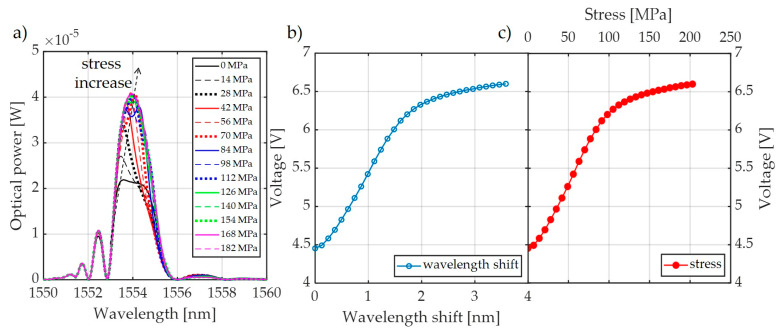
Measurement results using the interrogator presented in Figure 3a for the FBG + FBGF3 matched gratings pair: (**a**) change in the shape of the system output signal; (**b**) wavelength shift processing characteristics; (**c**) characteristics of the power line conductor stress conversion to changes in the voltage on the photodetector.

**Figure 16 sensors-20-02652-f016:**
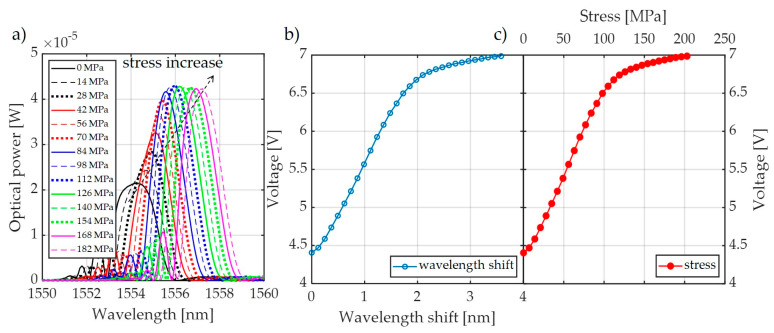
Measurement results using the interrogator presented in Figure 3b for the FBG + FBGF3 matched gratings pair: (**a**) change in the shape of the system output signal; (**b**) wavelength shift processing characteristics; (**c**) characteristics of the power line conductor stress conversion to changes in the voltage on the photodetector.

**Figure 17 sensors-20-02652-f017:**
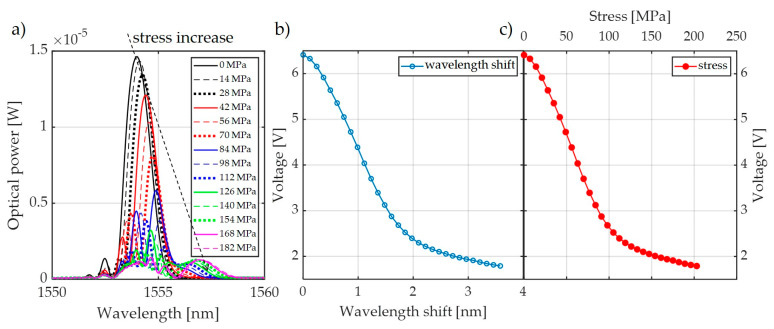
Measurement results using the interrogator presented in Figure 9 for the FBG + FBGF3 matched gratings pair: (**a**) change in the shape of the system output signal; (**b**) wavelength shift processing characteristics; (**c**) the characteristics of the power line conductor stress conversion to changes in the voltage on the photodetector.

**Figure 18 sensors-20-02652-f018:**
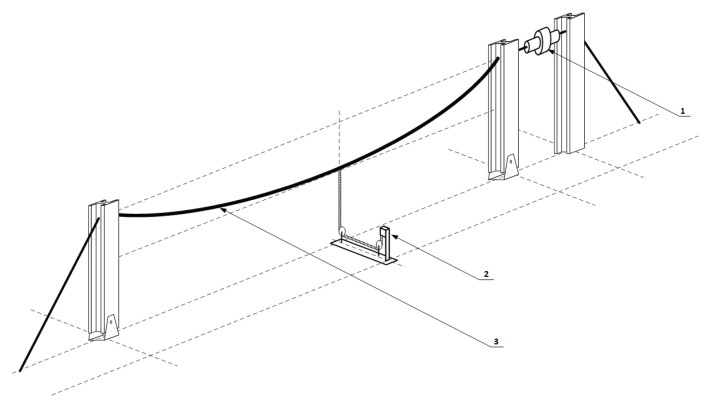
Outline of the outdoor testbed used for the proposed wire sag and temperature monitoring. 1: load cell Instron PM-L 2526-802 10 kN; 2: draw-wire displacement sensor Micro-Epsilon WDS 96/2500 mm; 3: ACSR Hawk wire with glued temperature sensor.

**Figure 19 sensors-20-02652-f019:**
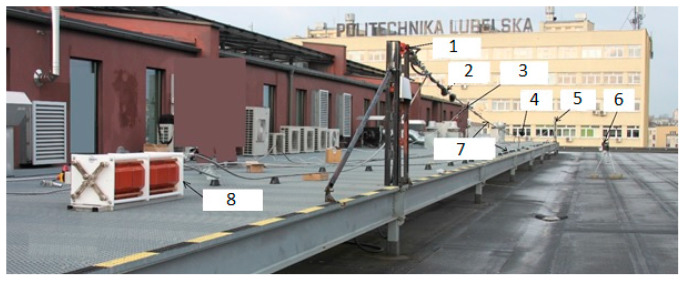
The outdoor test stand and its key elements. 1: Load cell; 2: proposed photonic sensor system; 3: temperature measurement; 4: sag measurement; 5: moving pole; 6: weather station; 7: ACSR Hawk conductor; 8: high-current transformer.

**Figure 20 sensors-20-02652-f020:**
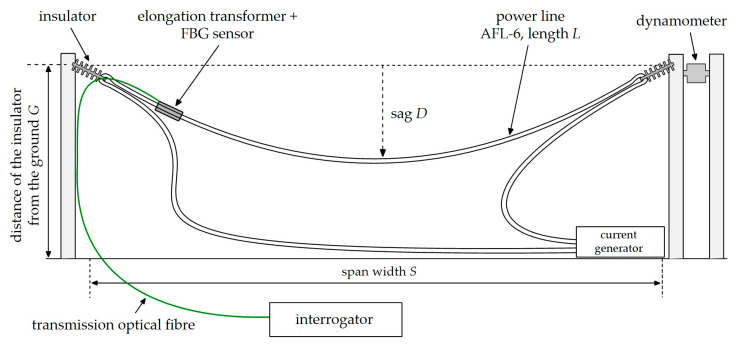
The overhead power line span at the outdoor testbed with the method of its characteristics’ parameters measurement.

**Figure 21 sensors-20-02652-f021:**
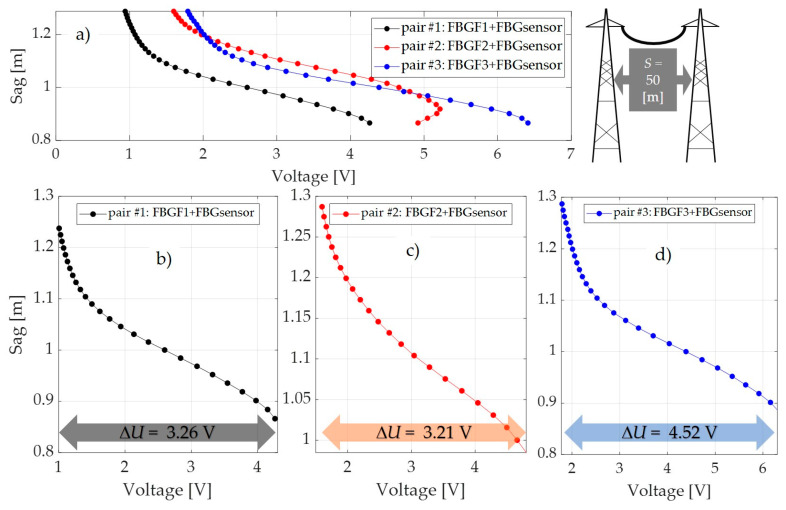
Results of power line sag value as a function of voltage measured by a photodetector for power line #1: (**a**) sag characteristics for three pairs of FBG structures; (**b**) sag dependence on the measured voltage on the photodetector for pair #1; (**c**) sag dependence on measured voltage on photodetector for pair #2; (**d**) sag dependence on measured voltage on photodetector for pair #3.

**Figure 22 sensors-20-02652-f022:**
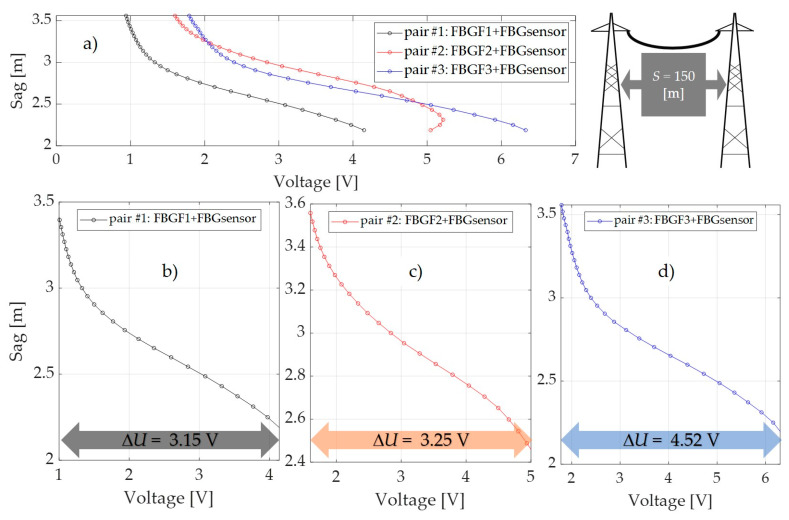
The results of the power line wire sag as a function of the photodetector measured voltages for power line #2: (**a**) the sag characteristics for three pairs of FBG structures; (**b**) sag dependence on the photodetector measured voltage for pair #1; (**c**) sag dependence on the photodetector measured voltage for pair #2; (**d**) sag dependence on the photodetector measured voltage for pair #3.

**Figure 23 sensors-20-02652-f023:**
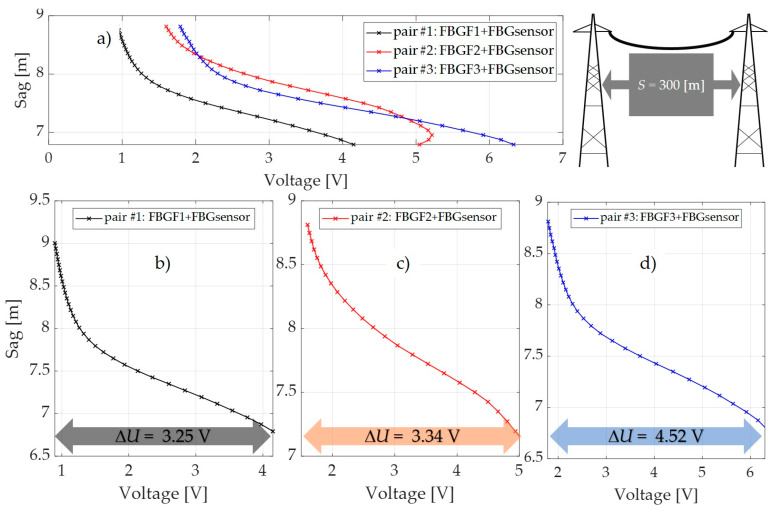
The results of the power line wire sag as a function of the photodetector measured voltages for power line # 3: (**a**) the sag characteristics for three pairs of FBG structures; (**b**) sag dependence on the photodetector measured voltage for pair #1; (**c**) sag dependence on the photodetector measured voltage for pair #2; (**d**) sag dependence on the photodetector measured voltage for pair #3.

**Table 1 sensors-20-02652-t001:** Parameters of tested spans.

Case	Wire Length (*L*)	Span Length (*S*)	Wire Cross-Section (*A*)	Initial Sag (*D_0_*)
#1	50.04 m	50 m	276.2 mm^2^	0.866 m
#2	150.08 m	150 m	276.2 mm^2^	2.121 m
#3	300.40 m	300 m	276.2 mm^2^	6.708 m

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
