# Peer review of "Overhead Transmission Line Sag Estimation Using the Simple Opto-Mechanical System with Fiber Bragg Gratings—Part 2: Interrogation System"

_sensors, 2020, doi:10.3390/s20092652_

Round 1

Reviewer 1 Report

This paper analyzes FBG interrogation schemes for overhead power lines monitoring particularly using matching FBGs.

- Is not clear in the paper what the novelty, considering that these interrogation schemes have been studied a lot. The authors must clarify this point.

- The authors declare in section 2:

 “ The response optical spectrum shape analysis of FBGs applied as the physical quantities transducers, causes difficulties because of optical spectrum analyzers or spectrophotometers requirements. Such devices are expensive, not portable and not capable of workin external, adverse environmental conditions”.

This is not entirely true, there are many portable FBG interrogators in the market that are portable and can work in adverse environmental conditions. A single example is:

- Considering the interrogation schemes presented in Figure 1, all the filters and FBG sensor are spatially separated. What is the proposal for temperature compensation, since the sensor and filter are placed in different temperatures and therefore the FBGs will be mismatched?

- In the Section 4 – Results, we can observe that the deformation curves are not linear. How the authors intend to correct this in terms of real deformation value? What is the accuracy of this method?

- It will be necessary a moderate English revision in the manuscript. A typical example is:

“Presented in the article researches, started from the installation of the entire elongation transformer system with the control plate and FBG sensor on the experimental outdoor teststand,  which was used for simulation of power line conductor operational conditions”

Reviewer 2 Report

References

10: Article title?

Line 96: Define SLED.

117: Isn’t it three types (a, b, and c) of interrogation systems?

211: Grammar “spectral characteristics of the sensor and filter cause that the processing characteristics are different”

219: Spelling “case”

238: Grammar “Figure 7 causes that the common”

242: Capitalize “Figure 7”

258: Capitalize “Fig. 7”

339: Spelling “mathematical”

Eqn. 1: Give a reference for where a reader may find this relation.

Introduction: Emphasize why it is important to measure the elongation of overhead power lines. Also, describe other technologies that accomplish this same goal and explain why Fiber Bragg Gratings are better.

Conclusion: State how your experiments could have been improved. Also, explain what other research should be investigated next to further this concept into actual implementation.

Round 2

Reviewer 1 Report

The authors answered the reviewer questions but all technical content of answers must be adapted in the manuscript text.

Author Response

All technical content of answers has been adapted in the manuscript text. The changes are marked with a blue background.